# Possible Causal Association between Type 2 Diabetes and Glycaemic Traits in Primary Open-Angle Glaucoma: A Two-Sample Mendelian Randomisation Study

**DOI:** 10.3390/biomedicines12040866

**Published:** 2024-04-15

**Authors:** Je Hyun Seo, Young Lee

**Affiliations:** 1Veterans Medical Research Institute, Veterans Health Service Medical Center, Seoul 05368, Republic of Korea; lyou7688@gmail.com; 2Department of Applied Statistics, Chung-Ang University, Seoul 06974, Republic of Korea

**Keywords:** primary open-angle glaucoma, mendelian randomisation, type 2 diabetes, fasting glucose, single-nucleotide polymorphisms

## Abstract

Existing literature suggests a controversial relationship between type 2 diabetes mellitus (T2D) and glaucoma. This study aimed to examine the potential causal connection between T2D and glycaemic traits (fasting glucose [FG] and glycated haemoglobin [HbA1c] levels) as exposures to primary open-angle glaucoma (POAG) in multi-ethnic populations. Single-nucleotide polymorphisms associated with exposure to T2D, FG, and HbA1c were selected as instrumental variables with significance (*p* < 5.0 × 10^−8^) from the genome-wide association study (GWAS)-based meta-analysis data available from the BioBank Japan and the UK Biobank (UKB). The GWAS for POAG was obtained from the meta-analyses of Genetic Epidemiology Research in Adult Health and Aging and the UKB. A two-sample Mendelian randomisation (MR) study was performed to assess the causal estimates using the inverse-variance weighted (IVW) method, and MR-Pleiotropy Residual Sum and Outlier test (MR–PRESSO). Significant causal associations of T2D (odds ratio [OR] = 1.05, 95% confidence interval [CI] = [1.00–1.10], *p* = 0.031 in IVW; OR = 1.06, 95% CI = [1.01–1.11], *p* = 0.017 in MR–PRESSO) and FG levels (OR = 1.19, 95% CI = [1.02–1.38], *p* = 0.026 in IVW; OR = 1.17, 95% CI = [1.01–1.35], *p* = 0.041 in MR–PRESSO) with POAG were observed, but not in HbA1c (all *p* > 0.05). The potential causal relationship between T2D or FG and POAG highlights its role in the prevention of POAG. Further investigation is necessary to authenticate these findings.

## 1. Introduction

Glaucoma is a major cause of permanent vision loss. It is a progressive condition that affects the optic nerve, leading to the deterioration of the retinal ganglion cells and their axons [1]. Primary open-angle glaucoma (POAG) is the predominant form of glaucoma subtype [2]; however, its pathogenesis remains unclear due to the role of multiple factors in its pathophysiology [3,4,5,6]. The proposed risk factors for glaucoma include ageing, elevated intraocular pressure (IOP), vascular factors, genetic factors, systemic disorders (such as diabetes), and environmental factors [3,5,6,7,8,9,10]. Thus, the identification of POAG causal risk factors may facilitate the early detection and prevention of glaucoma; therefore, these studies form the basis for eye and vision research.

Type 2 diabetes (T2D) is an increasingly prevalent chronic metabolic disorder [11,12] that affected approximately 415 million people in 2015 worldwide [13]. This representative systemic illness is frequently regarded as a systemic risk factor, along with systemic hypertension, for glaucoma prevention. However, in contrast to IOP and ageing in POAG, epidemiological findings regarding the effects of T2D on POAG development remain controversial [14,15,16,17,18]. The Blue Mountains Eye Study suggested a substantial correlation between T2D and POAG and considered it a risk factor [15]. Subsequently, several studies have examined the relationship between T2D and POAG, indicating that T2D may be a risk factor for POAG development with increasing IOP related to glycaemic traits [16,17,18]. However, the Rotterdam Study and Baltimore Eye Survey raised concerns regarding the non-significant association between T2D and POAG [19,20]. Additionally, recent studies have reported an insignificant association [21,22,23,24] or negative point estimate [20,25,26] between the two.

A large-scale study using the Korean National Health Insurance Data demonstrated that the hazard ratio of glaucoma for T2D was 1.80 (95% confidence interval [CI], 1.58–2.04) with adjustment [27]. Another meta-analysis suggested that upon comparing patients with and without diabetes, the pooled relative risk for glaucoma was 1.48 (95% CI, 1.29–1.71), with significant heterogeneity (I^2^ = 82.3%, *p* < 0.001) [28]. Due to this heterogeneity, it is unclear whether T2D is a risk factor for POAG. In addition, this retrospective association analysis was unable to prove the causality, thus, the nature of the association remains unclear.

Mendelian randomisation (MR) is a genetic epidemiological technique that employs genetic variants linked to potential exposures as the instrumental variables (IVs) to assess their causal impact on disease outcomes [29,30]. A previous study using MR analysis suggested variable evidence for an association between T2D and POAG (odds ratio [OR] = 1.97, 95% CI 1.01–1.15) in individuals with European ancestry [31]. However, a recent MR study of the Japanese population demonstrated that glycaemic traits such as fasting glucose (FG), glycated haemoglobin (HbA1c), and C-peptide levels did not display a significant correlation with POAG [32]. Although POAG prevalence differs between ethnic groups [7], it is a representative common complex disease in terms of genetics and multi-ethnic group analysis and is reliable if the subject pool is large enough for MR analysis [33,34]. Furthermore, a study on the two-sample MR analysis methodology using large cohorts, such as the UK Biobank (UKB), reported that the MR-Egger bias did not affect the inverse-variance weighted (IVW) and weighted median [35]. Moreover, the results of the MR analysis may vary based on the selection of IVs for T2D. Therefore, large datasets combining the meta-analysis of the Biobank Japan (BBJ) and UKB [36] are expected to generate more substantial results. To this end, this study aimed to conduct a two-sample MR analysis to investigate the possible causal effects of T2D and glycaemic traits (FG, and HbA1c levels) on POAG based on the BBJ and UKB meta-analyses [36], as well as the Genetic Epidemiology Research in Adult Health and Aging (GERA) and UKB meta-analyses [37].

## 2. Materials and Methods

### 2.1. Study Design

The study protocol was approved by the Institutional Review Board of the Veterans Health Service Medical Centre (IRB No. 2022-03-004), and the need for informed consent was waived because of its retrospective study design. The research was conducted in accordance with the tenets of the Declaration of Helsinki.

### 2.2. Data Sources

Figure 1 is a schematic of the analytical study design. To examine the potential causal effects of T2D and glycaemic traits (FG and HbA1c) on the risk of POAG, the following datasets were selected: (1) exposure data from the summary statistics of the genome-wide association study (GWAS)-based meta-analysis of the BBJ and UKB for the multi-ethnic population (*n* = 667,504 for T2D [84,224 cases vs. 583,280 controls], *n* = 448,252 for FG, and *n* = 415,403 for HbA1c) (Table 1) [36]; and (2) outcome data from the summary statistics of the POAG GWAS data from the meta-analysis (*n* = 240,302; [12,315 cases vs. 227,987 controls]) of the GERA and UKB [38]. POAG is defined by the International Classification of Diseases-9 diagnosis code of POAG or normal-tension glaucoma, excluding other subtypes of glaucoma (e.g., pseudoexfoliation, pigmentary, etc.) [38]. Table 1 enlists the datasets used for the summary statistics.

### 2.3. Selection of the Genetic IVs

Single-nucleotide polymorphisms (SNPs) associated with each exposure at the GWAS threshold *p* < 5.0 × 10^−8^ were used as IVs. To verify that each IV was independent of the other, the SNPs were pruned based on linkage disequilibrium (LD; r^2^ = 0.001, clumping distance = 10,000 kb). The 1000 Genomes Phase III Dataset (European population) was used as the reference panel to compute the LD for the clumping procedure. The F-value was determined using the formula *F* = *R*^2^(*n* − 2)/(1 − *R*^2^), where n is the sample size and *R*^2^ is the proportion of exposure variance by genetic variance [39]. F-values > 10 indicate the absence of a weak instrument bias [40].

### 2.4. Mendelian Randomisation

The MR analysis was conducted based on the following three presumptions concerning IVs: (1) they have to show a significant association with the exposure, (2) they must be unrelated to the confounding variables, and (3) they should solely affect the outcomes via exposure, indicating the absence of a directional horizontal pleiotropy effect. We employed the inverse variance-weighted (IVW) MR method with random effects as the major strategy [33,40,41]. The Cochran’s Q-test was used to evaluate the heterogeneity among SNPs in the IVW technique [41]. The presence of heterogeneity was shown by a *p*-value of less than 0.05 for Cochran’s Q-test. Heterogeneity may suggest the potential existence of horizontal pleiotropy. The effectiveness of IVW analysis is maximized when all genetic variations satisfy the three assumptions for IVs [42]. We conducted a sensitivity analysis to test the validity and reliability, taking into consideration potential concerns such as instrumental bias or pleiotropy. The weighted median approach [43], MR-Egger regression (with or without adjustment using the Simulation Extrapolation [SIMEX] method) [44,45], and the MR pleiotropy residual sum and outlier (MR-PRESSO) [46] were employed for sensitivity analysis. The weighted median approach yields reliable estimates, even when as many as 50% of the IVs are inaccurate [42]. The MR-Egger approach provides estimates of appropriate causal effects, even when pleiotropic effects are present, by taking into account a nonzero intercept that denotes the mean horizontal pleiotropic impacts and a slope that serves as an estimate of the causal impact [43]. If there is a violation of the assumption that there is no measurement error (I^2^ < 90%), bias can be addressed by employing MR-Egger regression with SIMEX [45]. The heterogeneity of the MR-Egger technique was assessed by the utilization of Rücker’s Q′ statistic tests [47]. The MR-PRESSO method is an expansion of the IVW with the objective of mitigating the presence of pleiotropic outliers [46]. The MR–PRESSO global test was employed to assess the presence of directional horizontal pleiotropy [46]. When the MR-PRESSO global test gives a *p*-value below 0.05, the MR-PRESSO outlier test is utilized to detect the presence of particular horizontal pleiotropic outlier variations [46]. As a consequence, the findings were interpreted in accordance with the suitable technique for MR analysis [48]. All analyses were conducted using the TwoSampleMR and SIMEX packages in R version 3.6.3 (R Core Team, Vienna, Austria).

## 3. Results

### 3.1. Genetic IVs

In total, 180 IVs were identified at the significance threshold values of *p* < 5.0 × 10^−8^ for T2D (Table 2). In addition, 108 and 303 IVs were identified at the significance limit of *p* < 5.0 × 10^−8^ for FG and HbA1c, respectively. The mean F-statistics for T2D, FG, and HbA1c (176.16, 111.30, and 119.61, respectively) used for MR were > 10, demonstrating a low likelihood of weak instrument bias (Table 2 and Appendix A). Detailed information on the IVs is provided in Appendix A.

### 3.2. Heterogeneity and Horizontal Pleiotropy of IVs

To evaluate the quality of the IVs, we computed the *I^2^* and *p* values for Cochran’s Q statistic using IVW, Rücker’s Q’ statistic using MR-Egger, and the MR-PRESSO global test, as displayed in Table 2. The Cochran’s Q test from IVW demonstrated that the IVs for T2D, FG, and HbA1c (all *p* < 0.001) were heterogeneous (Table 2); therefore, a random-effects IVW approach was used. Additionally, the Rücker’s Q′ test from the MR-Egger demonstrated heterogeneity between the IVs (all *p* < 0.001). Although heterogeneity suggests genetic variations could indicate pleiotropy, the MR-Egger regression intercepts did not show horizontal pleiotropy (*p* > 0.05) in all tests, regardless of the SIMEX correction (Table 2). In the MR-PRESSO global test for T2D, FG, and HbA1c, which showed substantial horizontal pleiotropic effects (all *p* < 0.001), the MR-PRESSO results were considered the primary outcomes based on prior research [48].

### 3.3. Mendelian Randomisation for the Possible Causal Association between T2D and POAG

T2D demonstrated a significant and probable causal association with glaucoma using the IVW method (MR OR = 1.05, 95% confidence interval (CI): 1.00–1.10 *p* = 0.031), weighted median method (MR OR = 1.08, 95% CI: 1.01–1.16, *p* = 0.026), and MR-PRESSO (MR OR = 1.06, 95% CI: 1.01–1.11 *p* = 0.017) (Figure 2). The genetic correlation between T2D and glaucoma for each SNP was a significant positive correlation in scatter plots (Figure 3).

### 3.4. Mendelian Randomisation for the Possible Causal Association of FG and HbA1c with POAG

FG demonstrated a significant causal association with POAG using the IVW method (MR OR = 1.19, 95% CI: 1.02–1.38 *p* = 0.026) and MR-PRESSO (MR OR = 1.17, 95% CI: 1.01–1.35, *p* = 0.041) (Figure 4). However, HbA1c did not demonstrate a significant causal association with POAG (all *p* > 0.05, all MR methods; Figure 4). Scatter plots indicate the genetic association between FG and HbA1c and that with POAG for each SNP (Figure 5).

## 4. Discussion

Our study demonstrated a possible causal association between T2D and POAG. Moreover, FG levels, which are popular glycaemic traits to diagnose T2D and prediabetes conditions, demonstrated a potential causal association with POAG. In contrast, HbA1c levels did not demonstrate a causal association with POAG.

Several observational studies have reported an association between T2D and glaucoma [15,49,50]. In addition, a meta-analysis has suggested that upon comparing individuals with and without diabetes, the pooled OR for POAG was 1.50 (95% CI, 1.16–1.93) [51]. However, several studies have reported an insignificant association [21,22,23,24] or negative point estimate [20,25,26]. Therefore, a large-scale study is required to address the disparities between these findings. A large meta-analysis, including 47 studies with 2,981,341 individuals, suggested that T2D is associated with POAG, indicating a pooled relative risk of 1.48 (95% CI: 1.29–1.71) [28]. In addition to an association, an MR analysis method was used to analyse these causal associations. Our study is consistent with the findings of an MR study, which reported on the possible causal relationship between POAG and T2D in Europeans (body mass index [BMI]-unadjusted: OR = 1.07, 95% CI, 1.01–1.14, and *p* = 0.028; BMI-adjusted: OR = 1.07, 95% CI, 1.01–1.15, and *p* = 0.035) [31] (Table 3). In our study, considering the possibility of pleiotropy due to the use of multi-ethnic genome-wide data, we conducted additional analyses using data composed of individuals of European descent (Additional File S1). As a result, we confirmed that T2D has a robust causal effect on POAG. The mechanistic consideration of the causality of T2D in POAG is necessary, and there is evidence from other studies that the presence of T2D causally contributes to an increase in IOP [52]. However, one previous study showed the possible causality between T2D and POAG was absent in the analysis of East Asian ancestry (BMI-unadjusted: OR = 1.01, 95% CI, 0.95–1.06, and *p* = 0.866; BMI-adjusted: OR = 1.00, 95% CI, 0.94–1.05, and *p* = 0.882) [31]. This difference can be attributed to the inclusion of approximately 46,000 East Asians in the outcome data, as well as the limited sample size, resulting in the possibility that the result may have been insignificant.

FG levels are often used for screening and evaluating prediabetes and T2D [54,55]. Elevated blood glucose levels, an important feature of T2D, are expected to be a reasonable indicator to evaluate the association between T2D and POAG. An observational study using 374,376 individuals from the Korea National Health Insurance data reported a positive association between FG levels and the incidence of glaucoma, with a hazard ratio of 2.022 (95% CI: 1.494–2.736) [56]. Similarly, we observed a strong association between FG and glaucoma using the MR analysis, which is a more stringent validation technique. Despite being a distinct genetic dataset, our results suggesting the causality of FG in POAG are substantial because they are novel and significant, compared with those of a previous study that used an MR analysis (Table 3). A hypothesis to explain this possibility may be that higher plasma FG is associated with higher glucose levels in the aqueous humour, which increases trabecular fibronectin levels and is associated with elevated IOP [57]. These hypotheses are supported by recent meta-analyses that suggest a pooled average increase of 0.09 mmHg in the IOP associated with a 10 mg/dL increase in the FG [28]. However, the association between diabetic retinopathy and glaucoma has been inconsistently demonstrated in several studies [58].

Clinically, HbA1c levels are associated with diabetic microvascular complications, which in turn are associated with long-term glycaemic control [59]. Researchers recommend maintaining a target HbA1c < 48 mmol/mol (6.5%) for the general population with T2D [60,61,62]. Regarding HbA1c and glucose levels, the Singapore Malay Eye Study demonstrated an elevated but insignificant trend, whereas a case-control study in Europe demonstrated a statistically significant association between elevated HbA1c levels and glaucoma [16]. However, HbA1c levels were not causally associated with POAG in our study, consistent with previous results (Table 3). Although we used 303 IVs in this study, heterogeneity and horizontal pleiotropy may have affected our results. An MR study using a large dataset demonstrated that HbA1c indicated marginal significance (*p* = 0.064); however, combined with the UKBB and FinnGen project dataset, the HbA1c indicated a possible causal association (OR: 1.28 95% CI, 1.01–1.61) [63]. A previous study had shown that the dose-response relationships between glucose metabolism markers and glaucoma prevalence are hockey-stick-shaped for HbA1c, and J-shaped for FG [18]. HbA1c quantifies glycaemic control over a period of 2 to 3 months, whereas FG assesses acute blood glucose levels. Consequently, FG is more sensitive to diseases as compared to HbA1c [64]. These different sensitivities in FG and HbA1C may lead to the different causal effects on POAG.

The chief strength of our study was the use of a relatively large cohort dataset, which suggested a possible causal association between T2D, and FG in glaucoma. However, this study had a few limitations. First, we did not have access to individual-level data; thus, we were unable to explain the presence of numerous confounding factors using summary statistics based on two-sample MR. Second, the test procedures to validate the MR hypotheses do not provide complete validation. The violations of MR assumptions can lead to invalid conclusions, thus warranting a cautious interpretation of the results. Third, few genome datasets include ophthalmic phenotype data; thus, it was difficult to separate and summarise a meta-analysis that included a portion of the UKB. However, considering the research results according to the large-cohort MR analysis methodology [35], the IVW and weighted median remain unaffected, which in turn influences the bias of MR-Egger. IVW and MR-PRESSO were the primary statistics in our study [48]; thus, the bias issue was minimised. In addition, there was no substantial difference between the MR methodologies, thus establishing the credibility of our results. Fourth, since our results contained heterogeneity issues, caution must be exercised when interpreting. The source of heterogeneity included the pleiotropy effect. As an alternative possibility, the samples used to estimate the SNP-exposure and SNP-outcome associations are not homogeneous; for example, a difference in the distribution of a covariate confounding the exposure-outcome relationship across samples could induce heterogeneity. In addition, the SNP-exposure and SNP-outcome relationships are not correctly specified—i.e., in the two-sample setting, the causal relationship between the exposure and the outcome is different in each of the samples [65]. Although we do not know the exact cause of heterogeneity, it was a multi-ethnic result, and since heterogeneity was not significant in the European race results that were additionally analysed (Additional File S1, [66]), it would be ideal to mention the possibility of heterogeneity due to heterogeneity in exposure and outcome data.

## 5. Conclusions

Our study demonstrated the possible causal association of T2D and FG on POAG development in European and East Asian populations using an MR analysis. The analysis of the European data set yielded consistent results, demonstrating the significance of POAG in T2D and enhancing the robustness and replicability of the findings. This potential causal relationship between T2D or FG and POAG highlights the significance of T2D in early detection and prevention of POAG, considering the high prevalence of T2D. Researchers should further clarify and investigate the association between T2D and POAG.

## Figures and Tables

**Figure 1 biomedicines-12-00866-f001:**
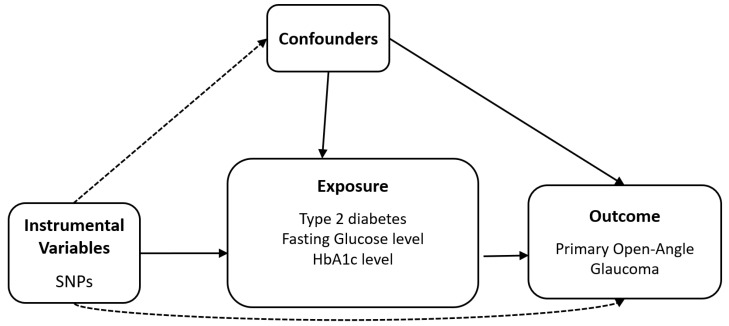
Diagram of two-sample Mendelian randomisation analysis. Abbreviation: HbA1c, glycated haemoglobin; SNP, Single nucleotide polymorphism.

**Figure 2 biomedicines-12-00866-f002:**
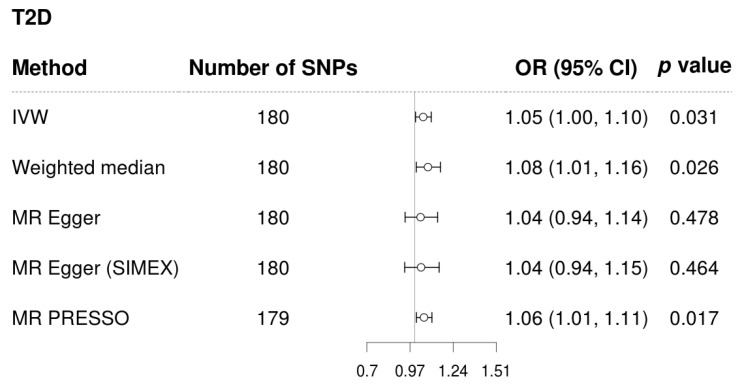
Forest plot of causal associations of T2D on glaucoma. Abbreviations: T2D, type 2 diabetes; IVW, inverse-variance weighted; SIMEX, Simulation Extrapolation; MR–PRESSO, MR- pleiotropy residual sum and outlier test; OR, odds ratio; CI, confidence interval.

**Figure 3 biomedicines-12-00866-f003:**
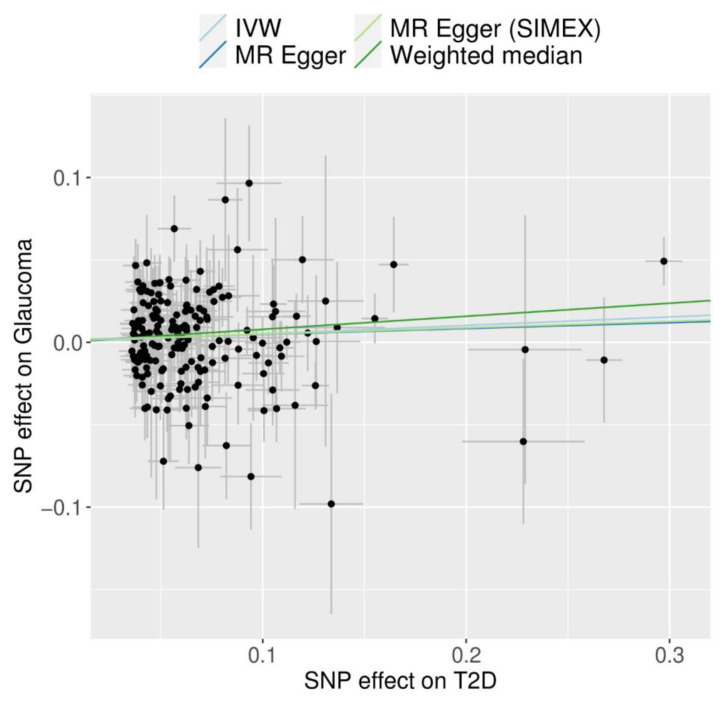
Scatter plots of MR tests assessing the effect of T2D on glaucoma. Abbreviations: T2D, type 2 diabetes; IVW, inverse-variance weighted; SIMEX, Simulation Extrapolation; MR, Mendelian randomisation. Light blue, light green, dark blue, and dark green regression lines represent the IVW, MR–Egger (SIMEX), MR–Egger, and weighted median estimate, respectively.

**Figure 4 biomedicines-12-00866-f004:**
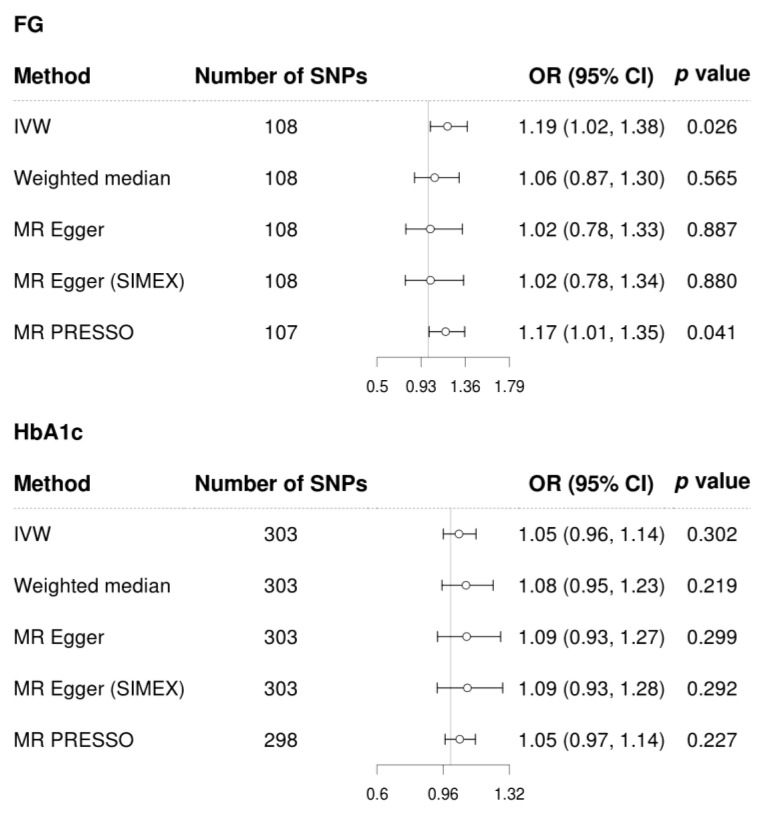
Forest plot of causal associations of FG and HbA1c on glaucoma. Abbreviations: FG, fasting glucose; IVW, inverse-variance weighted; SIMEX, Simulation Extrapolation; MR–PRESSO, MR-pleiotropy residual sum and outlier test; OR, odds ratio; CI, confidence interval, HbA1c, glycated haemoglobin.

**Figure 5 biomedicines-12-00866-f005:**
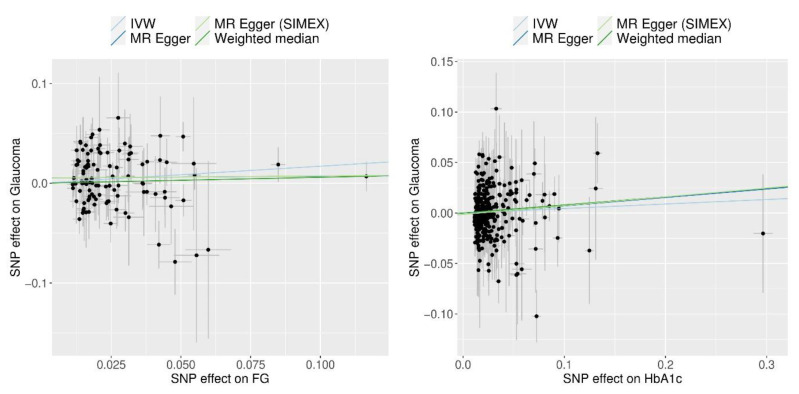
Scatter plots of MR tests assessing the effect of FG and HbA1c on glaucoma. Abbreviations: FG, fasting glucose; IVW, inverse-variance weighted; SIMEX, Simulation Extrapolation; HbA1c, glycated haemoglobin; MR, Mendelian randomisation. Light blue, light green, dark blue, and dark green regression lines represent the IVW, MR–Egger (SIMEX), MR–Egger, and weighted median estimate, respectively.

**Table 1 biomedicines-12-00866-t001:** Statistical measures summarizing the data source.

Traits	Data Source	Subjects Number	Population	Variants Number	Reference
T2D	BBJ Project + UKB	667,504 (84,224 cases + 583,280 controls)	East Asian + European	25,845,091	[36]
FG	BBJ Project + UKB	448,252	East Asian + European	20,535,873	[36]
HbA1c	BBJ Project + UKB	415,403	East Asian + European	20,525,742	[36]
Glaucoma	GERA cohort + UKB	240,302 (12,315 cases + 227,987 controls)	Multi-ethnic: 214,102 European 5103 African unspecified 3571 Other admixed ancestry 1847 African American or Afro-Caribbean 5189 Hispanic or Latin American 5370 East Asian 5120 South Asian	7,760,820	[37]

Abbreviations: T2D, type 2 diabetes; BBJ, BioBank Japan; UKB, UK Biobank; FG, fasting glucose; HbA1c, glycated haemoglobin; GERA, Genetic Epidemiology Research in Adult Health and Ageing.

**Table 2 biomedicines-12-00866-t002:** Heterogeneity and horizontal pleiotropy of instrumental variables.

Exposure				Heterogeneity		Horizontal Pleiotropy				
				Cochran’s Q Test from IVW	Rucker’s Q’ Test from MR-Egger	MR-PRESSO Global Test	MR-Egger		MR-Egger (SIMEX)	
	N	F	I^2^ (%)	*p*-Value	*p*-Value	*p*-Value	Intercept, β (SE)	*p*-Value	Intercept, β (SE)	*p*-Value
T2D	180	176.16	95.57	<0.001	<0.001	<0.001	0.001 (0.004)	0.720	0.001 (0.004)	0.771
FG	108	111.30	97.76	<0.001	<0.001	<0.001	0.005 (0.004)	0.179	0.005 (0.004)	0.191
HbA1c	303	119.61	97.63	<0.001	<0.001	<0.001	−0.001 (0.002)	0.565	−0.001 (0.002)	0.548

Abbreviation: N, number of instruments; F, mean F statistic; IVW, inverse-variance weighted; MR, Mendelian randomisation; PRESSO, pleiotropy residual sum and outlier; SIMEX, simulation extrapolation; β, beta coefficient; SE, standard error; T2D, type 2 diabetes; FG, fasting glucose; HbA1c, glycated haemoglobin.

**Table 3 biomedicines-12-00866-t003:** Comparison of previous studies using MR on type 2 diabetes and glycaemic traits on glaucoma.

Ethnicity	Exposure Dataset	Outcome Dataset	Instrumental Variables	Causal Association with Glaucoma	References
EUR	339,224	8591 cases, 210,201 controls	BMI: *n* = 64 WC: *n* = 36 WHR: *n* = 29	BMI: Significant WC: Significant WHR: NS	[52]
EUR	BMI: *n* = 339,224 WC and HC *n* = 224,459	1824 cases, 93,036 controls	BMI: *n* = 31 WC: *n* = 33 HC: *n* = 24	BMI: Significant WC: NS HC: Significant	[53]
EUR/EAS	T2D: EUR 74,124 cases, 824,006 controls EAS 77,418 cases, 356,122 controls FG and HbA1c EUR: 196,991 EAS: 36,584	182,702 EUR (15,229 cases, 177,473 controls)	T2D: *n* = 165 FG: *n* = 58 HbA1c: *n* = 60	T2D: Significant FG: NS HbA1c: NS	[31]
		46,523 EAS (6935 cases, 39,588 controls)	T2D: *n* = 129 FG: *n* = 11 HbA1c: *n* = 15	T2D: NS FG: NS HbA1c: NS	
EAS	FG: *n* = 17,289 HbA1c: *n* = 52,802 C-peptide: *n* = 1666	22,795 (3980 cases, 18,815 controls)	FG: *n* = 34 HbA1c: *n* = 43 C-peptide: *n* = 17	FG: NS HbA1c: NS C-peptide: NS	[32]
Multi-ethnicity	T2D: 667,504 FG: 448,252 HbA1c: 415,403	240,302 (12,315 cases, 227,987 controls)	T2D: *n* = 180 FG: *n* = 108 HbA1c: *n* = 303	T2D: Significant FG: Significant HbA1c: NS	This study

Abbreviations: EUR, Europeans; EAS, East Asians; BMI, body mass index; WC, waist circumference; WHR, waist hip ratio; HC, hip circumference; T2D, type 2 diabetes; FG, fasting glucose; HbA1c, glycated haemoglobin; NS, not significant.

## Data Availability

The datasets used and/or analysed in the current study are available from Biobank Japan (BBJ https://pheweb.jp/, accessed on 30 July 2022) [36] and the GWAS catalogue (https://www.ebi.ac.uk/gwas/summary-statistics, accessed on 19 July 2022).

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
