# Peer review of "Possible Causal Association between Type 2 Diabetes and Glycaemic Traits in Primary Open-Angle Glaucoma: A Two-Sample Mendelian Randomisation Study"

_biomedicines, 2024, doi:10.3390/biomedicines12040866_

Round 1

Reviewer 1 Report

Comments and Suggestions for Authors

This is a very interesting study that aims to elucidate the causal association between type 2 diabetes mellitus and primary open-angle glaucoma.

I would suggest the following modifications:

Could you please replace the phrase "causal association" with the phrase "possible causal association" in the entire text? The above replacement would be done in the title, as well.

Additionally, in line 264, HbA1c should be removed, as no correlation was found between primary open-angle glaucoma and HbA1c.

Author Response

Reviewer 1

This is a very interesting study that aims to elucidate the causal association between type 2 diabetes mellitus and primary open-angle glaucoma.

I would suggest the following modifications:

Could you please replace the phrase "causal association" with the phrase "possible causal association" in the entire text? The above replacement would be done in the title, as well.

Response: We sincerely thank you for your insightful suggestion! Since intraocular pressure is a major risk factor for glaucoma, we concur with this suggested revision. We have, accordingly, included the words “possible,” “probable,” and “potential” throughout the manuscript.

Additionally, in line 264, HbA1c should be removed, as no correlation was found between primary open-angle glaucoma and HbA1c.

Response: Thank you for your insightful suggestion. We have, accordingly, removed HbA1c from line 281.

Reviewer 2 Report

Comments and Suggestions for Authors

Clearly state the design anywhere it’s applicable

Principle of POAG diagnosing in the database used should be disclosed?
Table 2. 
What was a source of high heterogeneity?

230-245 not clear what unexpected in the fact that FG is associated with POAG, if DR established as an increased FG is associated with POAG. Same for IOP in 244-245 I doubt that this factor should be considered separately…

Author Response

Reviewer 2

Clearly state the design anywhere it’s applicable

Principle of POAG diagnosing in the database used should be disclosed?

Response: Thank you for your critical comment! The definition of POAG is a critical point in this glaucoma study. The summary statistics were defined as a prior study. We have also included a brief definition of POAG (lines 95-97).

Table 2. What was a source of high heterogeneity?

Response: Thank you for your thoughtful comment. As per Table 2, the p-values of Cochran’s Q, and Rucker’s Q’ test were <0.001 and indicate heterogeneity. The source of heterogeneity included the pleiotropy effect. As an alternative possibility, the samples used to estimate the SNP-exposure and SNP-outcome associations are not homogeneous; for example, a difference in the distribution of a covariate confounding the exposure-outcome relationship across samples could induce heterogeneity. Moreover, the SNP-exposure and SNP-outcome relationships are not correctly specified—i.e., in the two-sample setting, the causal relationship between the exposure and the outcome is different in each of the samples.1 Although we do not know the exact cause of heterogeneity, the study included multiple ethnicities, and since heterogeneity was not significant in the European race results that were additionally analyzed (Additional file 1), there is a possibility of heterogeneity due to heterogeneity in exposure and outcome data.

We have revised accordingly in the discussion section (lines 293—304; Additional file 1).

  1. Hemani G, Bowden J, Davey Smith G Evaluating the potential role of pleiotropy in Mendelian randomization studies Hum Mol Genet. 2018 Aug 1;27(R2):R195-R208. doi: 10.1093/hmg/ddy163.

230-245 not clear what unexpected in the fact that FG is associated with POAG, if DR established as an increased FG is associated with POAG.

Response: Thank you for your insightful comment. We have accordingly edited this statement for clarity and conciseness (lines 246-261).

Same for IOP in 244-245 I doubt that this factor should be considered separately…

Response: Thank you for your thoughtful comment. We apologize for the confusion and have edited the sentence for conciseness and clarity. We have also removed IOP from the same sentence (lines 255-261).

Reviewer 3 Report

Comments and Suggestions for Authors

Thanks for the opportunity to review the paper "Causal Association between Type 2 Diabetes and Glycaemic 2 Traits in Primary Open-Angle Glaucoma: A Two-Sample 3 Mendelian Randomisation Study". I appreciate the opportunity to contribute to the improvement of your work. The article investigates the potential causal relationship between type 2 diabetes and glycaemic traits as exposures to primary open-angle glaucoma in multi-ethnic populations. You use a two-sample Mendelian randomisation study to assess causal estimates.

I have read the article carefully and have some suggestions that I believe will help to improve its quality.

Overall, I believe that the article has potential, but it needs a thorough revision to improve the organization, clarity, and depth of the analysis. I recommend that you make the aforementioned changes to improve the quality of the article.

1. The plagiarism report indicates a percentage of 44%, which is considerably high. While it is understandable that some short phrases may coincide with previous work, section 2.4 appears to be almost completely plagiarized. I recommend that you carefully review this section and rewrite it in your own words to avoid plagiarism issues.

2. The conclusion of the abstract is not clear enough. It would be better if you could reformulate it to express the results and conclusions of the study more precisely.

3. The article lacks organization and coherence in the presentation of ideas. The sudden jumps between topics and the lack of a clear thread make the text difficult to understand. I recommend that you review the structure of the article and reorganize it so that the ideas flow logically and coherently.

4. The structure of the document is confusing. This lack of cohesion makes it difficult to understand and effectively transmit the message that you are trying to communicate. I recommend that you review the structure of the article and organize it so that the ideas flow logically and coherently.

5. The objectives stated at the beginning of the article are not clear enough, which makes it difficult to understand the intention and scope of the work. Similarly, the conclusions are not clearly supported by the data or arguments presented throughout the text, which generates a perception of weakness in the overall argument. I recommend that you review the objectives and conclusions to make them clearer and more precise.

6. The article does not present a critical analysis of the ideas presented or offer alternative perspectives. Possible objections or different approaches that enrich the discussion are not addressed, which limits the depth and breadth of the work. I recommend that you include a critical analysis that considers different perspectives and possible objections to strengthen the argument.

Author Response

Reviewer 3

Thanks for the opportunity to review the paper "Causal Association between Type 2 Diabetes and Glycaemic 2 Traits in Primary Open-Angle Glaucoma: A Two-Sample 3 Mendelian Randomisation Study". I appreciate the opportunity to contribute to the improvement of your work. The article investigates the potential causal relationship between type 2 diabetes and glycaemic traits as exposures to primary open-angle glaucoma in multi-ethnic populations. You use a two-sample Mendelian randomisation study to assess causal estimates.

I have read the article carefully and have some suggestions that I believe will help to improve its quality.

Overall, I believe that the article has potential, but it needs a thorough revision to improve the organization, clarity, and depth of the analysis. I recommend that you make the aforementioned changes to improve the quality of the article.

  1. The plagiarism report indicates a percentage of 44%, which is considerably high. While it is understandable that some short phrases may coincide with previous work, section 2.4 appears to be almost completely plagiarized. I recommend that you carefully review this section and rewrite it in your own words to avoid plagiarism issues.

Response: Thank you for this notification. We have revised the methods section to avoid plagiarism after reviewing for the same using the iTheniticate program. Since the format and methods used in the papers are similar, a high degree of similarity must be considered, and the highest similarity was measured at 5%. Entire similarity rate was less than 29% using iTheniticate.

  1. The conclusion of the abstract is not clear enough. It would be better if you could reformulate it to express the results and conclusions of the study more precisely.

Response: Thank you for your critical suggestion. Accordingly, we have edited the conclusion for conciseness. (lines 23-25).

  1. The article lacks organization and coherence in the presentation of ideas. The sudden jumps between topics and the lack of a clear thread make the text difficult to understand. I recommend that you review the structure of the article and reorganize it so that the ideas flow logically and coherently.

Response: Thank you for your critical suggestion. We have revised the manuscript accordingly for better readability.

  1. The structure of the document is confusing. This lack of cohesion makes it difficult to understand and effectively transmit the message that you are trying to communicate. I recommend that you review the structure of the article and organize it so that the ideas flow logically and coherently.

Response: Thank you for your critical suggestion. We have reviewed the article for readability. We have revised the manuscript once again and have used an English editing service (Editage) for better readability (lines 163-173; lines 246-261; lines 273-279).

  1. The objectives stated at the beginning of the article are not clear enough, which makes it difficult to understand the intention and scope of the work. Similarly, the conclusions are not clearly supported by the data or arguments presented throughout the text, which generates a perception of weakness in the overall argument. I recommend that you review the objectives and conclusions to make them clearer and more precise.

Response: Thank you for your critical suggestion. Accordingly, we have revised the same for clearer sentence structures and readability (lines 75-79; lines 306-313; Additional file 1).

  1. The article does not present a critical analysis of the ideas presented or offer alternative perspectives. Possible objections or different approaches that enrich the discussion are not addressed, which limits the depth and breadth of the work. I recommend that you include a critical analysis that considers different perspectives and possible objections to strengthen the argument

Response: Thank you for your critical comment. To enhance the significance of the study, we have performed an additional analysis for the enrichment of the study results, as well as a more comprehensive result (lines 273-279; lines 293-304; lines 306-313; and Additional file 1).